# Beam image-shift accelerated data acquisition for near-atomic resolution single-particle cryo-electron tomography

Jonathan Bouvette[1,8], Hsuan-Fu Liu[2,8], Xiaochen Du [3,4], Ye Zhou[3], Andrew P. Sikkema [5], Juliana da Fonseca Rezende e Mello[1], Bradley P. Klemm [1], Rick Huang[6], Roel M. Schaaper[1], Mario J. Borgnia [1✉] & Alberto Bartesaghi [2,3,7✉]

Tomographic reconstruction of cryopreserved specimens imaged in an electron microscope followed by extraction and averaging of sub-volumes has been successfully used to derive atomic models of macromolecules in their biological environment. Eliminating biochemical isolation steps required by other techniques, this method opens up the cell to in-situ structural studies. However, the need to compensate for errors in targeting introduced during mechanical navigation of the specimen significantly slows down tomographic data collection thus limiting its practical value. Here, we introduce protocols for tilt-series acquisition and processing that accelerate data collection speed by up to an order of magnitude and improve map resolution compared to existing approaches. We achieve this by using beam-image shift to multiply the number of areas imaged at each stage position, by integrating geometrical constraints during imaging to achieve high precision targeting, and by performing per-tilt astigmatic CTF estimation and data-driven exposure weighting to improve final map resolution. We validated our beam image-shift electron cryo-tomography (BISECT) approach by determining the structure of a low molecular weight target (~300 kDa) at 3.6 Å resolution where density for individual side chains is clearly resolved.

[1] Genome Integrity and Structural Biology Laboratory, National Institute of Environmental Health Sciences, National Institutes of Health, Department of Health and Human Services, Research Triangle Park, NC, USA. [2] Department of Biochemistry, Duke University School of Medicine, Durham, NC, USA. [3] Department of Computer Science, Duke University, Durham, NC, USA. [4] Department of Chemistry, Duke University, Durham, NC, USA. [5] Epigenetics & Stem Cell Biology Laboratory, National Institute of Environmental Health Sciences, National Institutes of Health, Department of Health and Human Services, Research Triangle Park, NC, USA. [6] Laboratory of Cell Biology, Center for Cancer Research, National Cancer Institute, National Institutes of Health, Department of Health and Human Services, Bethesda, MD, USA. [7] Department of Electrical and Computer Engineering, Duke University, Durham, NC, USA. [8] These authors contributed equally: Jonathan Bouvette, Hsuan-Fu Liu. ✉email: mario.borgnia2@nih.gov; alberto.bartesaghi@duke.edu

Three-dimensional cryo-electron microscopy (cryo-EM) is now routinely used to determine the structure of macromolecular complexes at near-atomic resolution. The method consists in reconstructing the electron scattering map of a macromolecule of interest from projection images of cryopreserved specimens recorded in a transmission electron microscope. Electron radiation sensitivity limits the dose that can be used and consequently the signal-to-noise ratio (SNR) of molecular images. Single-particle analysis (SPA) overcomes this limitation by averaging thousands of low-dose images of randomly oriented but otherwise identical macromolecules. Tens to hundreds of individual molecular projections are typically captured in a snapshot of a single field of view. The relative orientation of each individual particle projection can be determined provided that images are recorded over a uniform background and that the projections do not overlap. These requirements preclude the applicability of SPA to important questions in which the complex of interest cannot be isolated from its crowded and variable biological context. This limitation can be overcome using cryo-electron tomography (CET), where the electron scattering map or tomogram of an individual pleomorphic object is reconstructed from a series of images recorded while tilting the stage.

Multiple instances of the same biological entity can thus be identified within their native context, extracted from several tomograms and combined using sub-volume averaging (SVA) to obtain a high-resolution map[1–5]. The resolution of structures determined by SVA is generally lower than SPA, in part due to difficulties in estimating the contrast transfer function (CTF) accurately, the lack of effective strategies to compensate for radiation damage and the challenges of collecting large datasets due to the slow speed of data collection. The first sub-nanometer resolution structures determined by CET were obtained using the Constrained Single Particle Tomography (CSPT) approach[6], where the idea of working directly with the raw 2D projections extracted from the tilt-series and performing an SPA-like reconstruction was initially introduced. While the raw projections still carry information from the surrounding environment, image alignment based on the features of the repetitive structure will cause the variable signal from the background to be averaged out during 3D reconstruction. This hybrid strategy provides a convenient framework to analyze low contrast tomographic projections and produce high-resolution 3D maps by combining established principles for SPA image alignment and reconstruction, with enforcement of the tilt-geometry constraints. Despite these and other advances, however, fundamental technical barriers still exist that have limited the application of near-atomic resolution CET/SVA to either large molecular weight targets like ribosomes[7], or samples that display favorable spatial arrangements such as those present in quasi-crystalline formations of HIV-1 Gag polyprotein[8,9].

Protocols for data collection in cryo-EM consist of executing repetitive cycles of targeting and imaging. It follows that the rate of data collection is inversely proportional to the time it takes to position the beam on a target area and acquire an exposure. While the speed of image recording devices has improved significantly over the past decade and the trend is likely to continue, navigation between targets is still performed by mechanically displacing the specimen-holding stage using step motors. With few exceptions, the mechanical stages in most instruments have extremely limited in-plane precision (less than 200 nm) and poor eucentricity (~2 mm variation). While targeting errors and drift can be monitored and corrected using a tracking area, this strategy introduces additional steps that further reduce acquisition speed. In SPA, targeting errors in the plane are seldom critical because each area is imaged only once and errors in Z-height only result in small variations in defocus that are corrected during data processing. In CET, however, each target needs to be imaged from multiple angles while maintaining the object of interest within a narrow field of view, and defocus is difficult to estimate because of the low SNR images. Thus, precise correction of spatial parameters becomes a time-consuming requirement that is critical for improving resolution. Recently, the use of fast detectors has allowed the collection of continuous tilt-series or fast incremental single exposure (FISE) in as little as 5 min[10,11]. Part of the gain in speed is accomplished by skipping all real-time corrections while using a special high-eucentricity single-axis stage. Unfortunately, structures solved from data collected in this manner are poorly resolved compared to those resulting from more precisely corrected protocols for data collection.

When collecting data for SPA, navigating the local area electron-optically by using beam-tilt induced image-shift (BIS) is a fast and precise way to increase the number of targets acquired for each slow mechanical movement[12,13]. Extending this approach to tomography seems counterintuitive as targeting errors in a tilted specimen are exacerbated, thus requiring increased tracking precision. Here, we show that achieving precise targeting when expanding the surveyed area by electronic navigation requires additional modeling of the specimen behavior. This strategy increases the number of targets collected per stage movement, thereby accelerating data collection without sacrificing resolution. Moreover, this method can be implemented on instruments without specialized stages, and by virtue of limiting the number of mechanical movements required for navigation, it also reduces the wear on the stage.

To overcome the inherent low-throughput characteristic of CET data collection, improve the resolution of SVA and extend its application to a wider set of samples including low molecular weight targets, here, we: (1) use BIS navigation to multiply the number of regions of interest (ROI) imaged at each mechanical step, (2) integrate image processing into data collection to compensate the imprecision of the stage using geometrical constraints of multiplexed ROIs, and (3) introduce improved data processing routines for high-resolution structure determination using CET/SVA. Our beam image-shift electron cryo-tomography (BISECT) approach can produce tilt-series that simultaneously match the quality of data obtained by the dose-symmetric tilt-scheme[14], and the speed of FISE[10], effectively combining their benefits. During data processing, we improve map resolution by implementing accurate per-tilt astigmatic CTF estimation and self-tuning routines for exposure weighting during 3D reconstruction. To validate our approach, we collected tilt-series using BISECT and determined the 3.6 Å resolution structure of a low molecular weight target (~300 kDa), indicating that complexes within this size range are now within the reach of CET. The advances reported here considerably improve the technical capability of CET, bringing this technology a step closer to becoming a high-throughput tool for in-situ structure determination.

## Results

**Electronic navigation for multiplexed collection of neighboring ROIs.** Achieving sub-nanometer resolution in CET requires the combination of multiple tightly overlapping projections of each ROI recorded from different angles and with a well-defined defocus. Tracking areas are often used to estimate the deviations induced from mechanically tilting the sample and compensate them to ensure precise targeting. This time-consuming step can be turned into an advantage provided that the model used for compensating the errors is precise enough to enable targeting of adjacent ROIs by electronic means, thus dividing its cost by the number of multiplexed targets. To successfully implement this beam-image shift enhanced cryo-tomography, the movement of

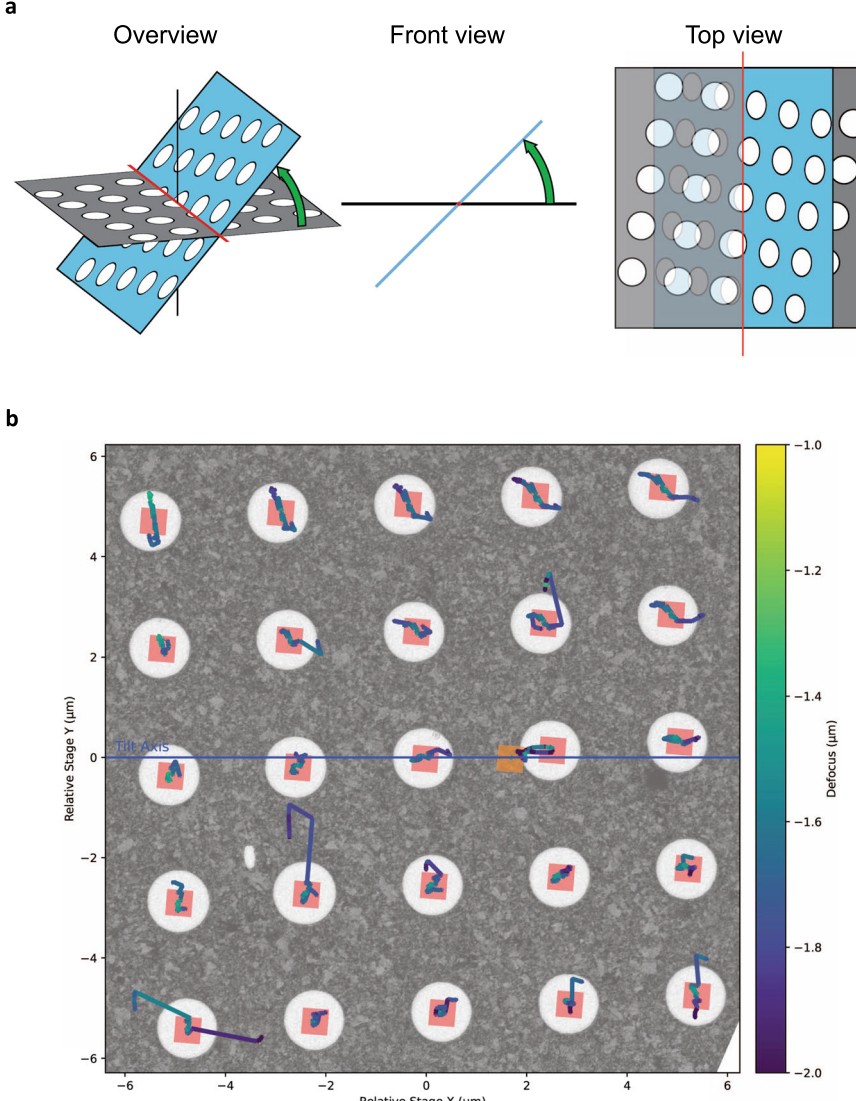

**Fig. 1 Parallel acquisition of tilt-series using beam-image shift electron cryo-tomography (BISECT). a** Schematic of the movement of ROIs (white circles) from the initial position (gray) and after stage tilting (blue). **b** Evolution of the targeting and defocus over a 5 × 5-hole area. The initial field of view of each area is shown as red squares. The high-magnification tracking area is shown as an orange square. Lines show the movement of the targeting obtained from cross-correlation alignment (scaled by 5 for visualization) and are color coded according to the estimated defocus for each image (large movements are attributed to errors in the cross-correlation at high-tilt and are not reflective of the actual motion of the sample). To allow for errors and discrepancies in the targeting of each hole, every BIS area is treated individually and gets its own Z-height fit. This approach makes our routines more robust to irregular specimens that strongly deviate from an ideal plane such as the bending observed in samples with thicker ice.

each ROI in all three dimensions needs to be considered. As the stage is tilted, targets that do not lie along the tilt axis will move away from the focus plane (Z-axis) and towards the tilt axis, Fig. 1a. Using partial tilt series alignment at each angle, we can estimate the new positions of each ROI from their previous locations. This iterative process contributes new tracking information to progressively improve the model, Supplementary Fig. 1a. Using this procedure, it is possible to set up image-shift following both regular holey grid pattern and arbitrary patterns where each ROI is manually selected, Fig. 1b. The actual number of target areas selected for imaging will determine the overall speed-up factor that can be achieved during acquisition, Supplementary Fig. 1b.

To satisfy the sampling rate requirements of imaging at near-atomic resolution, pixel sizes of 1.5 Å or smaller need to be used. At these magnifications, the field of view is rather small on the detector (≤600 nm on a K2 summit) while the precision of the

stage is in the same order of magnitude[11]. Without performing corrections between tilts, the targeted area would drift outside the field of view, especially when using a dose-symmetric scheme in which the stage is subject to large rotations[14]. To solve this problem, we adopted a two-step targeting strategy consisting of a coarse correction at low magnification followed by a second more accurate tracking step at the target magnification. This allows the acquisition of tilt-series at small pixel sizes without the need of a highly eucentric holder. As any resulting error in tracking is translated to all ROIs, we realized that achieving the necessary precision on the image-shift areas required an accuracy of <10 nm of the tracking area. The current implementation of tracking in SerialEM[15] correlates the current tracking image with the previous one on the same branch (positive or negative) of the tilt-series. Unfortunately, cross-correlation alignment of noisy images often leads to erroneous estimations of displacement. When shifts are calculated solely in relation to the last image from

the same branch, tracking errors accumulate over the course of data collection leading to significant deviations from the target. To improve precision and robustness, we calculated the displacement relative to the whole tracking series by iteratively aligning the cosine stretched images. To verify the correctness of the alignment, a new image is recorded after the calculated shifts are applied and the alignment process is repeated until a pre-established threshold is reached. Combined with the two-step tracking procedure (with thresholds of 100 nm and 5 nm at the low and high-magnification levels, respectively), this approach yields a tracking precision better than 10 nm over the entire course of the tilt-series, Supplementary Movie 1.

After implementing the tracking procedure described above and correcting for the image-shift of area locations using basic geometry relationships, we achieved a targeting precision that allowed successful tomogram reconstruction at high magnification. However, we observed that errors were inconsistent across different ROIs, resulting in many tilt-series moving off target. These errors in targeting also led to a wide spread of defocus across the tilt-series since the estimation of the vertical displacement of each ROI was dependent on a precise in-plane targeting. We realized that the basic geometric model was valid only when ROIs are co-planar with the focus/tracking area and that variations in the topography of the grid as small as 100 nm can be highly detrimental to targeting, Fig. 2a. By fitting axis rotation equations to the targeting data, we can get an initial estimation of the eucentric height for all ROIs using as few as the first five data points. From this fit, new image shift coordinates

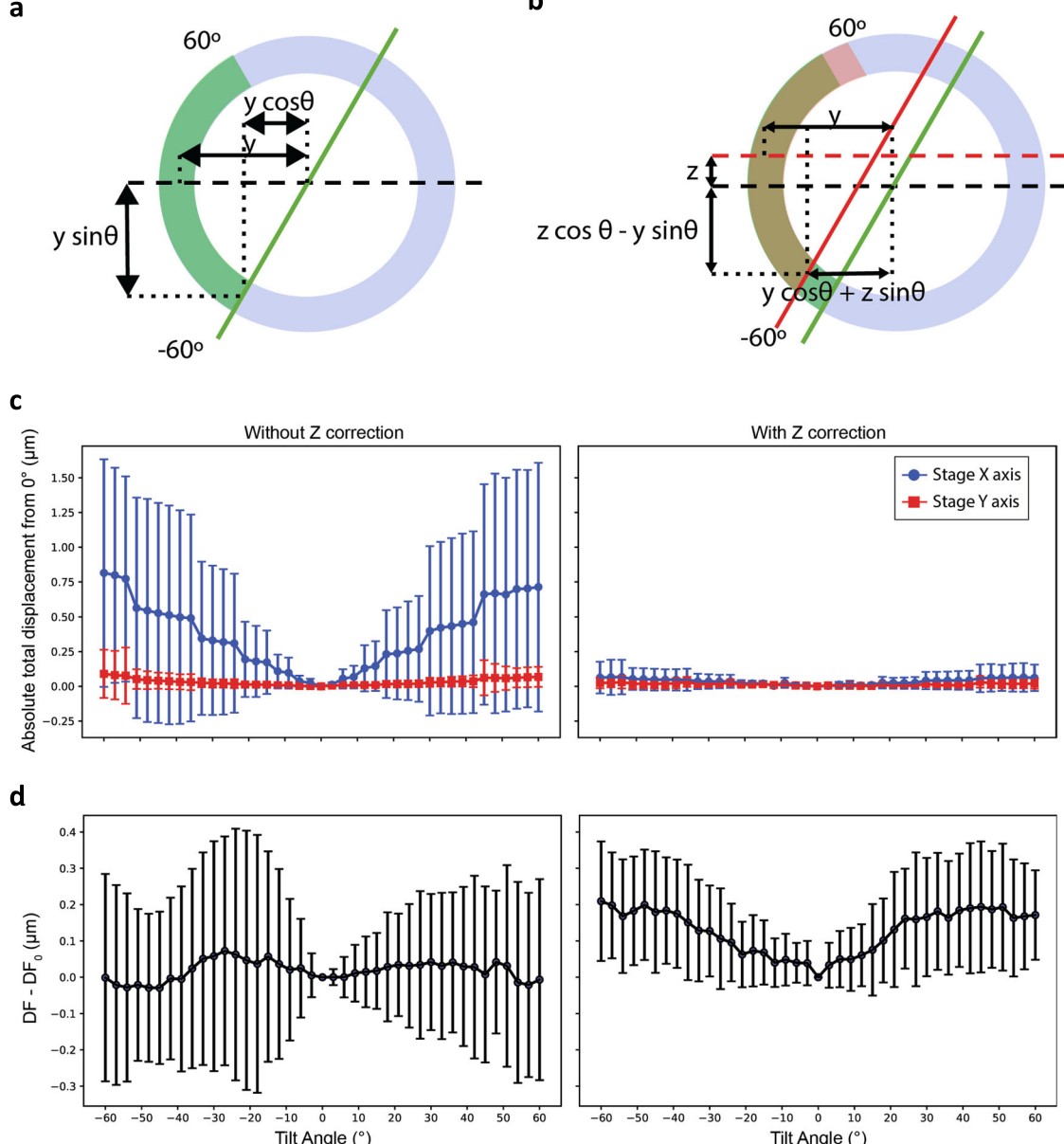

**Fig. 2 Effects of eucentric plane correction during tilt-series acquisition. a, b** Comparison of the target displacement during tilting assuming that the sample is flat compared to the center of rotation (**a**, green arc) or when considering an off-plane shift (**b**, red arc and red lines). Note that the trajectory of an off-plane target is different than for an eucentric target. **c** Measurement of the median displacement in the *x*-coordinate (blue) and *y*-coordinate (red) without correcting for off-plane shift (left panel) and with correction (right-panel) over multiple tilt-series (n=150). Error bars represent the inter quartile range (IQR).
**d** Measurement of the median defocus variation from the first image without correcting for off-plane shift (left panel) or with correction (right-panel) over multiple tilt-series (*n* = 150). Error bars represent the IQR. In general, errors in defocus originate mainly from errors in targeting and are not systematic.

and defocus corrections are calculated using the more accurate geometric relationships and the fit is refined with each additional point (Methods). This strategy minimizes the targeting errors and the defocus variations over the tilt-series, Fig. 2b, c. Generally, the targeting is within $0.067 \pm 0.124$ μm (median ± IQR, $n = 125$) and the defocus is within $0.21 \pm 0.17$ μm over the course of a tilt-series across image-shift areas up to the tested 8 μm distance from the tracking area (124 μm$^2$ total area, 25 holes on R1.2/1.3 holey grids). While the absence of coma aberration correction in tomography will likely prevent the use of much larger distances for high-resolution studies, targeting and defocus errors for each ROI bear no proportion to its BIS distance from the tracking area. In fact, we've observed that one of the most distal ROIs requiring a 7.2 μm BIS bore a near perfect targeting with a maximal in plane displacement of 68 nm and a variation in defocus of 0.48 μm across the entire tilt-series, Supplementary Movie 2. These results show that implementation of BISECT is feasible and that the current precision we achieve during targeting should allow for the use of even larger acquisition areas. For validation, we successfully imaged several CET samples with varying ice-thickness conditions and achieved between 2 and 15 min per tilt-series depending on the number of ROIs imaged per BIS area, the type of detector, tilt-range and number of tilts used during acquisition, Supplementary Table 1.

**Tilted-CTF determination from low-dose tomographic tilt-series.** Accurate defocus estimation is critical for correctly performing deconvolution of the CTF during 3D reconstruction and a prerequisite for achieving high-resolution. Small errors arising from imprecisions in the detection of zero-crossings can lead to signal attenuation, particularly at high-resolution. While strategies for fully astigmatic defocus estimation and correction are used routinely in SPA, their use in CET has been challenging for two main reasons: (1) the lower contrast of dose-fractionated tilted projections produces weak Thon rings that make defocus estimation unreliable, and (2) accounting for the defocus gradient induced by tilting requires the use of more sophisticated strategies for CTF estimation. To overcome these challenges, early attempts assumed eucentricity of the tilt-series and used iso-defocus strips running parallel to the tilt-axis to estimate defocus[16]. Other approaches either assume negligible astigmatism[17], estimate defocus for small subsets of projections in order to improve SNR[18], estimate per-tilt astigmatism but only use data from strips with minimal defocus variation (which can lead to inaccurate estimates especially at high-tilt angles)[19], or make regularizing assumptions about the progression of astigmatism in each tilt-series[20]. Meanwhile, strategies for collecting projections of tilted specimens were introduced in SPA for addressing problems of preferred orientation[21]. This led to the development of programs for tilted defocus determination that can estimate the orientation of the tilt-axis and the tilt angle based on local changes in estimated defocus[22–24]. While the lower doses used in tomography in principle prevent these strategies from working on tilt-series images, we reasoned that since the tilt-axis and tilt-angle for each projection are determined precisely during tilt-series alignment[25], these values could simply be used as input to one of the tilted CTF estimation routines used in SPA. This approach allowed us to reliably estimate defocus and astigmatism for each image in a tilt-series even at high-tilt angles, Fig. 3. We validated this strategy on tilt-series from *E. coli* 70S ribosomes (EMPIAR-10304), the SARS CoV-2 spike protein imaged on intact virions (EMPIAR-10453), and thin lamella generated by FIB milling from yeast cells (EMPIAR-10376), demonstrating its effectiveness on different types of CET samples and under a wide range of defocus and ice-thickness conditions, Supplementary Fig. 2.

**Constrained projection refinement and self-tuning exposure weighting.** As initially proposed in the CSPT approach[6], we carry out all 3D reconstruction and refinement operations using 2D particle projections extracted directly from the raw tilt-series. In addition, we improved the original CSPT algorithm by using newer components for 3D alignment and reconstruction to benefit from the latest algorithmic advances in SPA and implemented more accurate strategies for assigning per-particle CTF parameters based on the new astigmatic tilted-CTF model and the position of particles within tomograms, Supplementary Fig. 3. To maximize the recovery of high-resolution information during 3D reconstruction, we also implemented a new data-driven exposure weighting routine similar to the one we used to improve SPA reconstructions[26]. Exposure weights for each tilt angle are estimated based on the average similarity measured between raw particle projections and the most recent 3D reconstruction, Fig. 4a. The newly derived weights automatically reduce the contribution of high-tilt images or projections taken later in the tilt-series, allowing us to more efficiently extract high-resolution information from the raw tilt-series data, Fig. 4b. Since our strategy is purely data-driven, it requires no knowledge of the tilting scheme used during data collection and/or the electron exposure used for imaging, which can be difficult to determine accurately in practice. The use of data-driven strategies for exposure weighting has previously been shown to outperform approaches that use fixed critical exposure curves[27] both in SPA[26] and more recently in CET[7] (at least in the case of large or highly symmetric macromolecular complexes). Similar to the approach in[7], we also derive per-tilt weights based on the evaluation of similarities between particle images and reprojections of the 3D model, but we use the absolute value of the cross-correlation (not the signed version)[28] and isotropic B-factors to reduce overfitting. To validate our approach for high-resolution SVA, we re-processed tilt-series from several benchmark datasets including mammalian 80S ribosomes (EMPIAR-10064)[29], *E. coli* 70S ribosomes (EMPIAR-10304)[11], and the RyR1 ion channel imaged within native membranes (EMPIAR-10452)[30]. In all cases, we achieved resolutions that compare favorably with those obtained by other packages, showing the effectiveness of our approach when applied to large molecular weight targets (~2–3 MDa), Supplementary Figs. 4–6. In general, since our exposure weighting approach uses correlation scores derived from the alignment process, we expect this strategy to perform at least as well on other favorable targets such as those that display favorable spatial arrangements where the presence of symmetry facilitates image alignment[8].

**3.6 Å resolution structure of 300 kDa complex using BISECT/CSPT.** To determine the applicability of BISECT/CSPT to more challenging targets of smaller molecular weight, we imaged a monodisperse sample of protein VC1979 of *Vibrio cholerae* (dNTPase), an HD-domain family protein and a homolog of SAMHD1 and *E. coli* dGTP triphosphohydrolase. dNTPase is a homohexamer with D3 symmetry and a total molecular weight of ~300 kDa. We collected 11 areas of 25 holes for a total of 275 tilt-series using a K2 camera (~5 min per tilt-series). The on-the-fly cross-correlation tracking of the areas was within 220 nm for each position and the defocus estimation of individual images was within 0.4 μm, Fig. 2c, d. We estimated the tilted CTF for each projection independently and selected 64 tilt-series based on the absence of ice contaminants and the best estimated CTF resolution. Measuring the relative position of the selected ROIs with respect to the location of the tilt-axis revealed that good quality tilt-series were obtained from across the 5 × 5 BIS pattern, Supplementary Fig. 7. From the reconstructed tomograms, 34,435

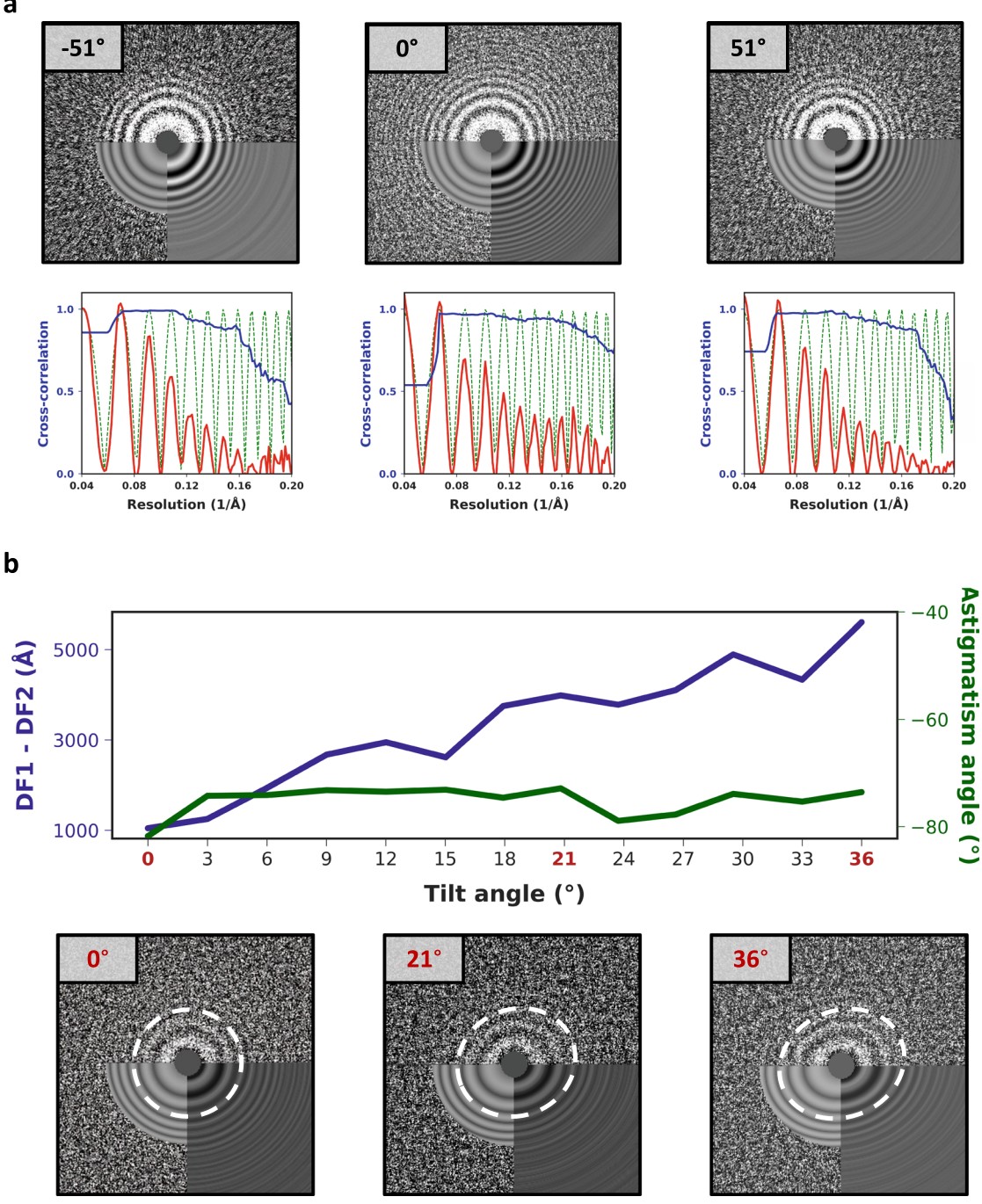

**Fig. 3 Astigmatic tilted-CTF determination from low-dose tomographic tilt-series. a** 2D power spectra from projections at −51°, 0°, and 51° from a tilt-series downloaded from EMPIAR-10453 (top) and corresponding fitted 1D models (bottom) showing radially averaged CTF profile (red), estimated defocus (green dashed line), and CTFfit scores (blue). **b** Astigmatic CTF estimation in the positive branch of a tilt-series from the dNTPase sample. Plots show variations in the astigmatism angle (green) and defocus differences (blue) across the tilt range (top). Corresponding 2D power spectra obtained from tilt angles 0, 21 and 36° showing changes in astigmatism (bottom). To facilitate the visualization, we choose to show a tilt-series with unusually high astigmatism variation. In general, no systematic variations in astigmatism were observed for data collected using BISECT or downloaded from the EMPIAR database.

particles were extracted and subjected to SVA followed by CSPT refinement, resulting in a 3.6 Å resolution map where density for side chains could be clearly visualized, Fig. 5 and Supplementary Fig. 8. This result validates the effectiveness of our combined approach for high-speed data collection and high-resolution CET/SVA data processing and represents the first example of a near-atomic resolution structure of a low-molecular weight target obtained from tomographic tilt-series. We also compared the

quality of our BISECT/CSPT structure against standard SPA reconstruction using images acquired from the same grid. We collected a single-particle dataset consisting of 2275 movies (using the same 25 holes BIS scheme) and extracted a similar number of particles as in the CSPT reconstruction (~34k) from the best 64 micrographs, yielding a reconstruction at 3.3 Å resolution using standard protocols for SPA refinement[26], Fig. 6a. The improvement in resolution is consistent with the higher quality of the SPA

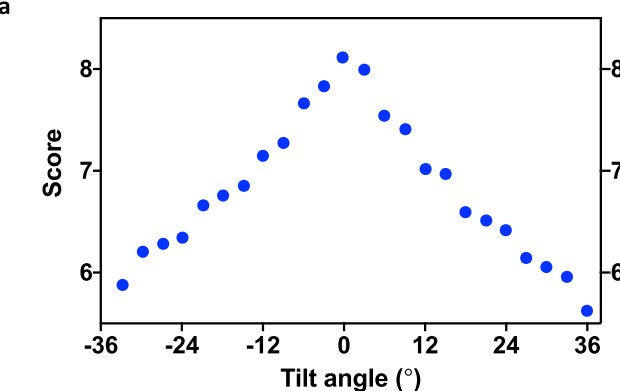

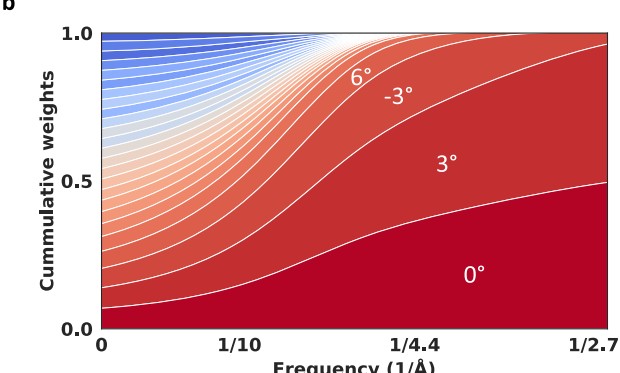

**Fig. 4 Self-tuning exposure weighting based on assigned particle scores.**
**a** Average similarity scores calculated over individual particle projections and plotted as a function of the tilt-angle revealing the expected characteristic of the dose-symmetric scheme showing higher quality data obtained at the lower tilts. **b** 2D plot showing the cumulative contribution of individual tilts to the 3D reconstruction as a function of spatial frequency. Each colored band represents one tilt. The high frequency information is contributed mostly by lower tilts which are also exposed first.

images resulting from choosing only the top 3% of movies (64 out of 2275) to produce the final map (vs. 64 out of 275 tilt-series that were used for the CSPT reconstruction). In addition, the cumulative effects of beam-induced motion are likely more pronounced in tomography due to the use of multiple shorter exposures (as opposed to a single long exposure in SPA), also contributing to the lower resolution. Despite the reduced overall quality of the CSPT reconstruction, both maps show similar density for individual side chains in the ordered regions of the structure, Fig. 6 and Supplementary Fig. 9, suggesting that under otherwise equal experimental conditions, BISECT combined with CSPT can produce maps of comparable quality to SPA. Moreover, the use of multiple tilted projections in certain situations (e.g., samples displaying extreme conformational heterogeneity and/or heavy orientation bias) could potentially lead to improvements in the accuracy of particle alignments and therefore to better resolutions.

## Discussion

While the number of SPA structures deposited in the Electron Microscopy Data Bank (EMDB) continues to increase, progress in CET has been slower due in part to the technical challenges involved in imaging crowded specimens using low-dose settings. Here, we present a platform for tilt-series data collection and processing that significantly improves the technical capability of CET, paving the way for the routine visualization of targets

in-situ at near-atomic resolution. By simultaneously achieving high-quality and high-speed data acquisition in combination with effective strategies for data processing, our methods can streamline the determination of high-resolution structures and provide a path for imaging biological targets present at low concentrations within their native environments which are not tractable by current low-throughput methods.

Our approach extends the single tilt BIS strategy widely used in SPA to the case of tilted data collection for tomography. This is accomplished by modeling the behavior of the specimen in real time to increase the accuracy of electronic navigation to multiple areas. In the current implementation, we use a separate computer to calculate the corrections in real time. The results are then transmitted to the data collection software to maintain the target within the field of view of the detector. This configuration is adaptable to existing imaging setups with virtually no modification. A powerful alternative would be to perform these calculations directly at the processing unit of the detector and send the appropriate corrections to the microscope computer. This would allow the detector to lock onto the field of view for each ROI. The resulting improvement in speed may make it possible to combine BIS multiplexed data collection with FISE without sacrificing resolution.

Using single-particle tomography, we have shown that BISECT can accelerate tomographic data collection while preserving high-resolution information. The question remains on whether it is applicable to cellular tomography. The most promising approaches for in-situ structure determination involve preparation of frozen hydrated subcellular slices either by serial sectioning[31–33] or by focused ion beam (FIB) milling[34,35]. Target areas, often identified using fluorescent markers, are too large to be sampled at atomic resolution with current detectors. Although sampling could be extended by acquiring consecutive tilt-series by mechanical displacement of the specimen, this approach would be too slow to be practical. The ability of BISECT to image an arbitrary pattern of areas in a radius of approximately 8 μm, allowing parallel collection of several tilt series spanning the size of an average mammalian cell, effectively provides a path for overcoming this limitation. Moreover, when compared to serial acquisition by mechanical displacement, BISECT requires a significantly lower number of tracking images, thus reducing the overall radiation damage that is likely to destabilize the specimen. Cases where a single feature of interest is present within the BIS range will not benefit from using our approach, however, since the permissive distance of 8 μm is consistent with the average size of a mammalian cell, we expect most samples to have multiple ROIs within this radius and therefore benefit from using BISECT.

Our strategy for data processing capitalizes on recent advances in SPA by repurposing these tools to work on low-dose projections extracted from tomographic tilt-series. Imposition of the constraints of the tilt-geometry during refinement and reconstruction allows us to accurately align particle projections and correct for the CTF while minimizing overfitting. While we currently cannot assess the effects of beam-tilt in the resolution of the dNTPase structure (due to the low contrast of the individual particle projections), we expect these effects to be similar in magnitude to those observed in SPA[12,13]. In the future, extension of beam-tilt correction strategies currently used in SPA[28] to CET could result in further improvements in map quality. This and future developments in data processing will result in additional improvements in resolution that could soon allow the application of these methods to even lower molecular weight targets (<300 kDa).

While SPA has been the flagship modality of cryo-EM to determine structures of proteins in monodisperse state at ever increasing resolutions, the advent of technological advances in

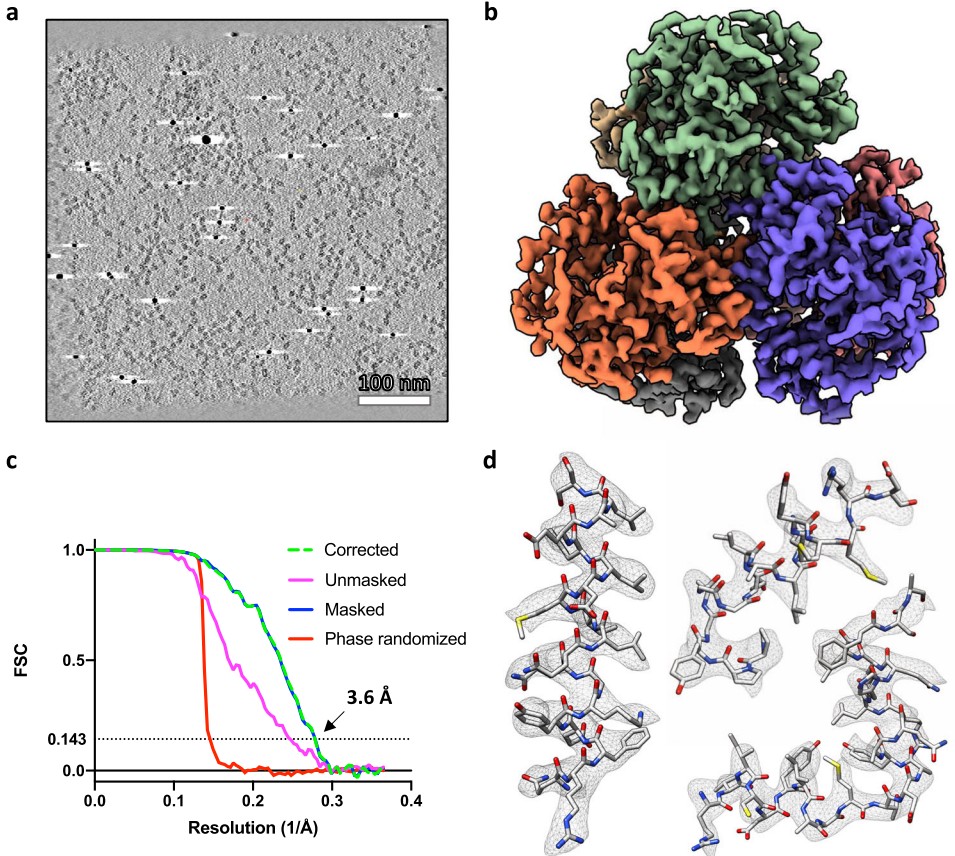

**Fig. 5 Near-atomic resolution structure of 300 kDa complex using BISECT/CSPT. a** Representative 50 nm-thick slice through tomographic reconstruction showing individual dNTPase particles (a total of 64 tomograms were used). Scale bar 100 nm. **b** Overview of dNTPase map obtained from ~30k particles extracted from 64 low-dose tomographic tilt-series. **c** Fourier Shell Correlation plots between half-maps showing an estimated resolution of 3.6 Å according to the 0.143-cutoff criteria. Curves corrected for masking effects (green dashed line), unmasked (magenta), masked (blue), and phase randomized (red) are shown. **d** Regions of the density map and fitted atomic coordinates showing the visualization of side chains. 3D refinement and reconstruction were repeated independently three times yielding structures with similar resolution.

CET like the ones reported here will pave the way for this technique to become an effective strategy for routinely studying protein complexes at near-atomic resolution within the functional context of the cell. Importantly, at these resolutions, the visualization of individual side chains greatly facilitates the placement of atomic models into the cryo-EM maps, resulting in more accurate structures. Ultimately, these methods will help close the resolution gap between high-resolution strategies used to study molecular assemblies reconstituted in vitro like X-ray crystallography and single-particle cryo-EM, and techniques for in-situ structure determination like CET.

## Methods

**Sample preparation**. dNTPase was cloned into pMCSG7 using LIC cloning, Supplementary Table 3. It was then transformed in BL21(DE3) and grown in Terrific Broth (TB) medium with 4% glycerol, ampicillin (100 μg/ml). Cultures were induced with IPTG (final concentration 0.2 mM), and grown for an additional 14–18 h at 20 °C before harvesting by centrifugation. Cell pellets were resuspended in approximately 5 mL Buffer A (100 mM Tris pH 7.9, 500 mM NaCl, 30 mM imidazole, 5% glycerol) per 1 g of cell paste. All purification steps were performed at 4 °C. Resuspended cells were incubated with DNase (4 mg), lysozyme (40 mg), and MgCl$_2$ (2 mM). Cells were lysed by three 1 min sonication cycles at 60% power using a Branson 450 Digital Sonifier with at least 2 min between cycles and clarified by centrifugation at 30,000$g$ for 30 min. The soluble fractions were loaded onto a 5-mL HisTrap Ni-NTA column (GE Healthcare) in Buffer A and eluted with an imidazole gradient from 45 to 300 mM. Sample was dialyzed overnight into Buffer B (50 mM Tris pH 7.9, 100 mM NaCl, 5% glycerol) at 4 °C and treated with tobacco etch virus (TEV) protease (1:30 ratio TEV:dNTPase plus 2 mM DTT) to remove the purification (polyhistidine) tag, and then applied to a 5-mL HisTrap Ni NTA column. The flow-through fraction was further purified by size exclusion

chromatography with a HiLoad 16/600 Superdex 200 column (GE Healthcare) in Buffer C (50 mM Tris pH 7.9, 100 mM NaCl, 10% glycerol). Peak fractions were pooled, concentrated to 10–20 mg/mL, aliquoted, flash cooled with liquid N$_2$, and stored at −80 °C.

**Grid preparation**. 1–1.5 mg protein was thawed then run over a 10/30 Superdex 200 Increase column (GE Healthcare) in Buffer D (20 mM Tris pH 7.5, 100 mM NaCl). 2 parts of the peak fraction at about 1.3 mg/mL was mixed of with 1 part of 6 nm gold fiducials that were 20x concentrated and buffer exchanged in Buffer D. 3uL of the mix was applied on a UltrAuFoil R1.2/1.3 300 mesh grid (Quantifoil Micro Tools GmbH). Grids were previously glow-discharged for 30 sec at 15 mA (PELCO easiGlow™, Ted Pella, Inc.). The excess sample was removed by blotting for 3 s using filter paper (Whatman #1) and grids were plunged-frozen in liquid ethane (−182 °C) using a vitrification robot (EM GP2, Leica Microsystems).

**dNTPase data collection**. Data were collected at 300 keV under parallel beam illumination on a Titan Krios (Thermo Fisher Scientific) housed at the NCI cryoEM facility affiliated with the NIH IRP CryoEM Consortium (NICE). Images were recorded on a direct electron detector (K2 summit, Gatan Inc.) placed behind an energy filter (Bioquantum, Gatan Inc.) operated with a 20 eV slit. Tilt-series were collected using SerialEM (version 3.8beta8)[15]. An initial set of 125 tilt-series were collected using a ±60° range grouped dose-symmetric scheme (0, 3, −3, −6, 6, 9, −9, −12, 12, 15, −15, −18, −21, 18, 21, 24, 27, −24, −27, −30, −33, 30, 33, 36, 39, 42, −36, −39, −42, −45, −48, 45, 48, 51, 54, 57, 60, −51, −54, −57, −60) with a dose of ~3e⁻/Å² per tilt angle (4 frames per movie, ~120 e⁻/Å² total dose). An additional 275 tilt-series were collected using a similar tilt scheme as before but with a ± 36° range (0, 3, −3, −6, 6, 9, −9, −12, 12, 15, −15, −18, −21, 18, 21, 24, −24, −27, −30, 27, 30, 33, 36, −33, −36) and a dose of ~5e⁻/Å² per tilt angle (11 frames per movie, ~120 e⁻/Å² total dose). Additional CET data collection statistics are included in Supplementary Table 1. The SPA dataset was collected using a total dose of ~60 e⁻/Å². For all three datasets, each area was collected with a pixel size of 1.37 Å/pix using a 5 × 5-hole lattice with the stage centered on the target area and

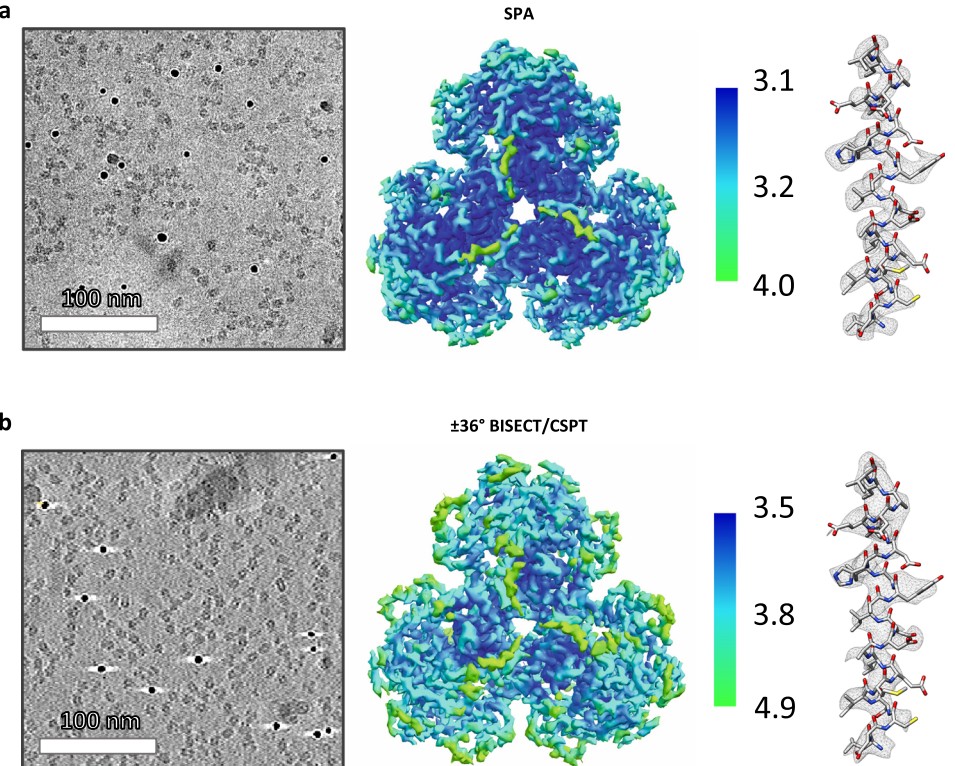

**Fig. 6 Comparison of SPA and CET structures of dNTPase. a** Representative 2D micrograph selected out of a total of 64 images (left, scale bar 100 nm), SPA map colored according to local resolution (middle) and corresponding density for an alpha helix (right). **b** Representative slice through tomographic reconstruction selected out of a total of 64 tomograms (left, scale bar 100 nm), ±36° BISECT/CSPT map colored according to local resolution (middle), and corresponding density for an alpha helix (right). 3D refinement and reconstruction were repeated independently three times yielding structures with similar resolution.

each hole acquired using BIS. The two CET datasets were used for structure determination by SVA and CSPT and the single-particle dataset was subjected to SPA refinement, Supplementary Table 2.

**BISECT protocol for tilt-series data collection.** A set of routines written in Python were used externally to track the tilt-series and correct for the image-shift coordinates and defocus positions. Tracking was done in 2 steps: (1) a low-magnification tracking step at 5.2 Å/pix with a threshold 100 nm, and (2) a high-magnification tracking step with a threshold of 5 nm. tiltxcorr[15,25], was used to calculate the global displacement of each tilt with respect to the initial image. Tracking was corrected with image shift and iterated until the thresholds above were met (the high-magnification threshold was usually attained within 1 or 2 iterations). To ensure robust tracking, the low mag threshold should be smaller than half the size of the field-of-view at the target magnification and the high mag threshold should be small enough to minimize the mismatch between tilts but sufficiently large to minimize the number of iterations needed for convergence. These thresholds need to be adjusted when changing the target magnification by making them proportional to the size of the new field-of-view. Targeting corrections for the ROIs also used tiltxcorr to calculate the alignments. The Z-height estimation was done by applying a 2D rotation to the $y$-coordinate of the target. After fitting for $z$, the new values of $y$ and defocus ($z$) were calculated using Eqs. (1) and (2):

$$y\prime = y \cos\theta + z \sin\theta \qquad (1)$$

$$z\prime = -y \sin\theta + z \cos\theta \qquad (2)$$

where $\theta$ is the tilt angle in degrees. CTF estimation of each individual image during acquisition was estimated with CTFFIND4[36]. The autofocusing routine uses the same magnification used for data collection and the focus area is shifted along the orientation of the tilt axis. BISECT's tracking strategy is different from that implemented in ref. [14]. Instead of applying a single alignment step, we iterate until a specified threshold is satisfied. Another difference is that the dose-symmetric scheme correlates each new image against the previous tilt in the branch while we compare against the first image of the tilt series (to prevent the accumulation of tracking errors). Similar to BIS strategies used in single-particle cryo-EM, our approach can also be used to target multiple ROIs per hole to further accelerate data collection speed. Timing statistics for data collected on a pleomorphic sample

of virus-like particles (VLPs) using R2/4 grids and four shots per hole are presented in Supplementary Table 1.

**Data collection times.** The speed of data collection varied with the type of microscope and detector used. Benchmarks were done using a 25-hole (5 × 5 lattice) BIS pattern, and a dose-symmetric scheme running at ±60° with 3° increments and 3 e−/Å²/tilt. On a Talos Arctica equipped with a K2 detector, we could achieve 8.1 min per tilt-series. Similarly, on a Titan Krios equipped with a K2 detector we achieved 7.6 min per tilt-series. Finally, on a Titan Krios equipped with a K3 detector, we could achieve 3.0 min per tilt-series. The rate-limiting step is the tracking routine, since the microscope has to wait for the alignments to return before starting the next step, while the alignment of the image-shift areas can occur before the start of the next cycle. The speeds achieved by BISECT are similar to those achieved by the FISE data collection scheme[10,11], and up to ten times faster than the dose-symmetric scheme[14].

**Determination of tilted-CTF model from tomographic tilt-series.** In order to estimate the defocus gradient of individual tilted projections, we first performed tilt-series alignment using routines implemented in the program IMOD[25]. The orientation of the tilt-axis and the tilt-angle for each projection were used as input to an updated version of the program CTFFIND4[36] that is part of the latest development version of cisTEM[28], available at https://github.com/ngrigorieff/cisTEM. It includes a new feature to determine the tilt angle and tilt axis in images of tilted specimens. The original code was adapted by simply disabling the search for the tilt-angle and tilt-axis orientation, and instead using the values determined during tilt-series alignment to calculate a single corrected power spectrum. Using this approach, we could accurately estimate the three CTF parameters for each projection in the tilt-series, Fig. 3 and Supplementary Fig. 2.

**3D refinement and data-driven exposure weighting.** After tomogram reconstruction, a low-resolution initial model was obtained using SVA, and particle projections from all sub-volumes were subsequently extracted and merged into one large particle stack while keeping track of the particle identities, parameters of the tilt-geometry and the per-particle defocus, Supplementary Fig. 3. Sub-volume orientations and the parameters of the tilt-geometry parameters were then refined using CSPT. Similarity scores were assigned to individual particle projections using the refine3d routine implemented in the cisTEM package[28]. To determine the

weights of the exposure filter, score averages over all particles in a tilt-series were calculated and used inside a modified version of cisTEM's reconstruct3d program. As expected, the measured score averages successfully captured the relative differences in image quality characteristic of the dose-symmetric tilt-scheme and the expected effects of radiation damage, Fig. 4a. The corresponding 2D frequency weights were derived using the same formula we implemented for exposure filtering in SPA[26], Fig. 4b.

**CSPT refinement of EMPIAR-10064, EMPIAR-10304, and EMPIAR-10452**. Tilt-series were aligned and reconstructed using routines implemented in IMOD and astigmatic per-tilt CTF models were estimated using the newly proposed routines. Sub-volumes were extracted either manually or using the coordinates provided in the EMPIAR entry and subjected to SVA using EMAN2 followed by CSPT refinement. Shape masks were used during refinement to down weight the contribution of the surrounding solvent area. For EMPIAR-10064, we obtained a resolution of 5.6 Å from 3,202 particles (compared to 8.6 Å obtained using emClarity[37], 8.4 Å obtained using EMAN2[17], and 5.7 Å obtained using M[7]), Supplementary Fig. 4. For EMPIAR-10304, we obtained a resolution of 4.8 Å from 10,279 particles (compared to 7 Å obtained using emClarity, EMD-10211[37]), Supplementary Fig. 5. For EMPIAR-10452, we obtained a map at 8.0 Å resolution from 2,715 particles (compared to 9.1 Å obtained using dyn2rel, EMD-10840[30]), Supplementary Fig. 6. In all cases, resolutions were estimated using the Fourier Shell Correlation (FSC) between half-maps using the 0.143-cutoff criteria after correcting for mask effects.

**Near-atomic resolution structure of dNTPase**. The set of 275 raw tilt-series was first subjected to the standard CET processing pipeline including tilt-series alignment and reconstruction as implemented in the package IMOD[25]. Astigmatic tilted-CTF models were determined for each tilted projection and the 64 tilt-series containing the highest resolution signal were selected for further processing. 34,435 particles were manually selected from the tomographic reconstructions and subjected to SVA and classification as implemented in the EMAN2 package[17], resulting in a 6.4 Å resolution reconstruction. Particle projections were then extracted from the raw tilt-series and defocus values assigned based on the position of each particle within the tomogram. 3D reconstruction was obtained using standard SPA routines implemented in cisTEM[28], followed by constrained refinement of particle orientations and parameters of the tilt-geometry using CSPT[6] resulting in a reconstruction at 3.6 Å resolution. Following a similar protocol, we also processed the first set of 125 raw-tilt series collected using a wider ±60° tilt range and obtained a reconstruction at 4.9 Å resolution from a similar number of particles, Supplementary Fig. 9. Additional data processing statistics are included in Supplementary Table 2. The lower resolution achieved from the ±60° dataset is a consequence of the weaker image contrast in each tilted projection due to fractionation of the dose across more images (41 vs. 25), which resulted in reduced accuracy of CTF estimation and less accurate image alignments. Final map resolution was estimated using the Fourier Shell Correlation between half-maps (0.143-cutoff criteria) after correcting for masking effects. For post-processing, maps were sharpened with phenix.resolve_cryo_em using as inputs the unfiltered half-maps, the molecular weight of the complex, and a soft shape mask[38]. An atomic model from X-ray crystallography was fit into the cryo-EM maps for visualization.

**Reporting summary**. Further information on experimental design is available in the Nature Research Reporting Summary linked to this paper.

## Data availability
Cryo-EM density maps of dNTPase determined using regular SPA and BISECT/CSPT have been deposited with the Electron Microscopy Data Bank (EMDB) with accession codes EMD-23355 and EMD-23356, respectively. The structure of the 80S ribosome obtained from EMPIAR-10064 has been deposited with accession code EMD-23357 and the structure of the E. coli 70S ribosome from EMPIAR-10304 with accession code EMD-23358. The atomic model of dNTPase is available from the Protein Data Bank (PDB) with accession code PDB ID 7LWZ. Other data are available from the corresponding authors upon reasonable request.

## Code availability
The SerialEM and Python scripts used to implement BISECT are available at https://gitlab.cs.duke.edu/bartesaghilab/bisect.

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

## Acknowledgements

The authors thank Qinwen Huang for assistance with data processing and James Horng for his role in the protein expression and purification. This work was supported in part by the Intramural Research Program of the NIH; National Institute of Environmental Health Sciences (ZIC ES103326 to M.J.B. and A.B. and ZO1 ES101905 to R.M.S.). This study utilized the NIH RIP cryo-EM facility (NICE) and the computational resources offered by Duke Research Computing (http://rc.duke.edu). The work was performed in part at the Duke University Shared Materials Instrumentation Facility (SMIF), a member of the North Carolina Research Triangle Nanotechnology Network (RTNN), which is supported by the National Science Foundation (award number ECCS-2025064) as part of the National Nanotechnology Coordinated Infrastructure (NNCI). We thank Tim Grant and Niko Grigorieff for making the development version of CTFFIND4 available to us and Mark DeLong, Charley Kneifel, Mike Newton, Victor Orlikowski, Tom Milledge, and David Lane from the Duke Office of Information Technology and Research Computing for providing assistance with the computing environment.

## Author contributions

J.B. and M.J.B. developed and implemented BISECT, H.L., X.D. and A.B. developed and implemented CSPT, A.P.S., B.K., and R.M.S. developed the V. cholerae VC1979 project, including the cloning and purification of the protein, J.B. and J.F.R.M. prepared specimens, J.B. and R.H. collected data, J.B., H.L., Y.Z., and A.B. did the data analysis, J.B., H.L., M.J.B., and A.B. wrote the paper, J.B., H.L., Y.Z., and A.B. prepared the figures, and M.J.B. and A.B. conceived the project.

## Competing interests

The authors declare no competing interests.
