## [Peer Review File · Nature Communications]

EDITORIAL NOTE: This manuscript has been previously reviewed at another journal that is not operating a transparent peer review scheme. This document only contains reviewer comments and rebuttal letters for versions considered at *Nature Communications*.

REVIEWERS' COMMENTS

Reviewer #1 (Remarks to the Author):

In the revised manuscript, the authors have resolved the problems and answered the questions I pointed out in my first review. I don't have further questions on this revised manuscript.

Minor comment:

In the Section "Near-atomic resolution structure of dNTPase" of ONLINE METHODS, the author added the results from two datasets (41 tilts per ROI, $\pm 60^\circ$ vs. 25 tilts per ROI, $\pm 36^\circ$) and comment on the differences in resolution (4.9 Å vs. 3.6 Å) and choice of parameters for data collection using BISECT/CSPT. It is mentioned that "at least in the case of monodisperse samples of small molecular-weight targets, this indicates that the use of BISECT using smaller tilt-ranges and fewer tilts (instead of the typical $\pm 60^\circ$ range with 3° increments) should be used if the goal is to achieve high resolution". If the author wants to draw such a conclusion, the data collection parameters for these two datasets should be the same except the tilt-ranges/number of tilts, and the numbers of sub-volumes/particles should be close in the image processing (29,925 vs. 34,435 in Supplementary Table 2). According to the Supplementary Table 2, the first dataset (41 tilts per ROI, $\pm 60^\circ$) has 4 frames per tilt, but the second dataset (25 tilts per ROI, $\pm 36^\circ$) has 11 frames per tilt. Assuming the total dose and dose rate are the same, the first dataset should have longer exposure time per frame than the second dataset. Longer exposure time per frame may also affect the high-resolution information.

Reviewer #2 (Remarks to the Author):

This is a good study and authors have responded appropriately to the reviewer comments. I have very minor comments:

1. Suggest to tone down the statement in the abstract "Here, we introduce protocols for tilt-series acquisition and processing that accelerate data collection speed by an order of magnitude and significantly improve map resolution compared to existing approaches" because the acceleration of data collection speed depends upon the sample (can be zero) and because against the most state-of-the-art approach (M), the demonstrated resolution improvement is 0.1 Å.
2. Update published biorxiv reference.

Reviewer #3 (Remarks to the Author):

I have read again the revised manuscript titled "Beam image-shift accelerated data acquisition for near-atomic resolution single-particle cryo-electron tomography" and found that the authors have implemented most of the recommendations from the referees. I particularly appreciated the new Supplementary Figure 4 that compares CSPT to other

approaches using the mammalian 80S ribosome dataset. Although I think an additional experiment on a biological sample not amenable to single particle analysis would increase the impact of the manuscript, the presented work is convincing enough within the stated scope that the BISECT/CSPT approach is able to preserve high resolution information at higher data collection speeds. I recommend that the editors consider this work for publication in Nature Communications.

Below are some minor comments:

The authors state “the movement of each ROI in all three dimensions needs to be considered” and “we can estimate the new positions of each ROI from their previous location”. One aspect of the BISECT workflow could be better clarified: How does tracking at a single location, which produces one displacement value each in X/Y at each tilt, result in different corrections for the various ROIs (as seen in Figure 1B)? Do these adjustments occur during the asynchronous processing step when partial tilt series are aligned?

The authors state “the precision of the stage is in the same order of magnitude (200 nm for a Titan Krios)”. Was this value measured by the authors, the manufacturer, or has it been reported before?

The authors write “Using this procedure, it is possible to set up image-shift following both regular and arbitrary patterns”. Some details on the patterns mentioned would help clarify this sentence.

In Supplementary Figure 1A, it is not clear how the box labelled “Asynchronous [sic] processing” feeds back into the workflow. Also, what do “Update report figure” and “Update summary report” refer to?

REVIEWERS' COMMENTS

Reviewer #1 (Remarks to the Author):

In the revised manuscript, the authors have resolved the problems and answered the questions I pointed out in my first review. I don't have further questions on this revised manuscript.

Minor comment:

In the Section "Near-atomic resolution structure of dNTPase" of ONLINE METHODS, the author added the results from two datasets (41 tilts per ROI, $\pm 60^\circ$ vs. 25 tilts per ROI, $\pm 36^\circ$) and comment on the differences in resolution (4.9 Å vs. 3.6 Å) and choice of parameters for data collection using BISECT/CSPT. It is mentioned that "at least in the case of monodisperse samples of small molecular-weight targets, this indicates that the use of BISECT using smaller tilt-ranges and fewer tilts (instead of the typical $\pm 60^\circ$ range with 3° increments) should be used if the goal is to achieve high resolution". If the author wants to draw such a conclusion, the data collection parameters for these two datasets should be the same except the tilt-ranges/number of tilts, and the numbers of sub-volumes/particles should be close in the image processing (29,925 vs. 34,435 in Supplementary Table 2). According to the Supplementary Table 2, the first dataset (41 tilts per ROI, $\pm 60^\circ$) has 4 frames per tilt, but the second dataset (25 tilts per ROI, $\pm 36^\circ$) has 11 frames per tilt. Assuming the total dose and dose rate are the same, the first dataset should have longer exposure time per frame than the second dataset. Longer exposure time per frame may also affect the high-resolution information.

We edited this statement and removed the conclusion cited by the reviewer to avoid any confusion.

Reviewer #2 (Remarks to the Author):

This is a good study and authors have responded appropriately to the reviewer comments. I have very minor comments:

1. Suggest to tone down the statement in the abstract "Here, we introduce protocols for tilt-series acquisition and processing that accelerate data collection speed by an order of magnitude and significantly improve map resolution compared to existing approaches" because the acceleration of data collection speed depends upon the sample (can be zero) and because against the most state-of-the-art approach (M), the demonstrated resolution improvement is 0.1 Å.

We edited the abstract to say: "that accelerate data collection speed by up to an order of magnitude" and removed "significantly" from the resolution improvement claim.

2. Update published biorxiv reference.

We updated this reference.

Reviewer #3 (Remarks to the Author):

I have read again the revised manuscript titled "Beam image-shift accelerated data acquisition for near-atomic resolution single-particle cryo-electron tomography" and found that the authors have implemented most of the recommendations from the referees. I particularly appreciated the new Supplementary Figure 4 that compares CSPT to other approaches using the mammalian 80S ribosome dataset. Although I think an additional experiment on a biological sample not amenable to single particle analysis would increase the impact of the manuscript, the presented work is convincing enough within the stated scope that the BISECT/CSPT approach is able to preserve high resolution information at higher data collection speeds. I recommend that the editors consider this work for publication in Nature Communications.

Below are some minor comments:

The authors state "the movement of each ROI in all three dimensions needs to be considered" and "we can estimate the new positions of each ROI from their previous location". One aspect of the BISECT workflow could be better clarified: How does tracking at a single location, which produces one displacement value each in X/Y at each tilt, result in different corrections for the various ROIs (as seen in Figure 1B)? Do these adjustments occur during the asynchronous processing step when partial tilt series are aligned?

We added the following statement to clarify this point: "Using partial tilt series alignment at each angle, we can estimate the new positions of each ROI from their previous locations. This iterative process contributes new tracking information to progressively improve the model, Supplementary Figure 1A".

The authors state "the precision of the stage is in the same order of magnitude (200 nm for a Titan Krios)". Was this value measured by the authors, the manufacturer, or has it been reported before?

Values in the same order of magnitude have been reported in the literature. We've removed the text "(200 nm for a Titan Krios)" and added the corresponding citation instead.

The authors write "Using this procedure, it is possible to set up image-shift following both regular and arbitrary patterns". Some details on the patterns mentioned would help clarify this sentence.

We edited this sentence to say: "Using this procedure, it is possible to set up image-shift following both regular holey grid pattern and arbitrary patterns where each ROI is manually selected".

In Supplementary Figure 1A, it is not clear how the box labelled "Asynchronous [sic] processing" feeds back into the workflow. Also, what do "Update report figure" and "Update summary report" refer to?

We updated this figure and now indicate that the async processing sends values back into the workflow and also clarified that the reports are updated into figures saved as png files.